# A New Sensing Material Based on Tetraaza/SBA15 for Rapid Detection of Copper(II) Ion in Water

**DOI:** 10.3390/membranes12111152

**Published:** 2022-11-16

**Authors:** Eda Yuhana-Ariffin, Siti Syahraini Sulaiman, Noraisyah Abdul Kadir Jilani, Devika Nokarajoo, Nurul Hidayah Abdul Razak, Darfizzi Derawi, Siti Aishah Hasbullah

**Affiliations:** Department of Chemical Sciences, Faculty of Science and Technology, Universiti Kebangsaan Malaysia (UKM), Bangi 43600, Selangor Darul Ehsan, Malaysia

**Keywords:** TL/SBA15, Cu^2+^ ion, optical sensor, tetraaza compound, reflectance measurement

## Abstract

A novel rapid and sensitive optical sensor for Cu^2+^ ion detection based on 5,5,7,12,12,14-hexamethyl-1,4,8,11-tetraazacyclotetradeca-7,14-dienium dibromide (TL) immobilized on Santa Barbara Amorphous (SBA-15) has been successfully developed. The inner and outer space of SBA15 allowed a high capacity of TL compound to immobilize onto it. FESEM (Field Emission Scanning Electron Microscopy) analysis was performed to confirm the morphology of TL-SBA15, while FTIR (Fourier Transform Infrared Spectroscopy) was utilized to confirm the interaction of TL–SBA15. A binding study of TL compound towards Cu^2+^ ion was performed via UV-vis solution study and binding titration. The stoichiometric binding ratio and binding constant value K*_b_* of TL towards Cu^2+^ ion was 1:1 and 2.33 × 10^3^ M^−1^, respectively. The optical reflectance sensor based on the TL compound is selective to Cu^2+^ ion and demonstrated a linear response over a Cu^2+^ ion concentration range of 1 × 10^−7^ M to 2 × 10^−5^ M, with a detection limit (LOD) of 1.02 × 10^−7^ M (R^2^ = 0.99) and fast response time of < 1 min. It showed high reproducibility, with a relative standard deviation (RSD) obtained at 0.47%. This optical sensor is reusable up to five consecutive times on Cu^2+^ ion by using 0.1 M EDTA with a pH of 6 as a regeneration solution, with a reversibility RSD value of 0.79%. The developed optical sensor provides a rapid and sensitive tool for Cu^2+^ ion detection in teabag samples, and the results align with those obtained by the ICP-MS standard method.

## 1. Introduction

70% of the Earth’s surface is covered with water. Only 0.5% is readily available for human use, and the rest comprises salt water (97%) and frozen fresh water (2.5%), either as polar ice or stored as ground water [1]. Water plays a major role in human life. All of mankind uses water for drinking and domestic purposes like cooking, washing, industrial processes, and agricultural activities. Furthermore, water is the biggest component of matter in the human body. For an average young adult male, total body water represents 50% to 70% of body weight [2]. The National Research Council (NRC) has recommended a daily water intake of 1 mL/kcal of energy expended. The Third National Health and Nutrition Examination Surveys (NHANES) obtained a result of total daily water intake for children and adults, and it indicates about 80% of total daily water intake is from beverages and 20% from food. The NHANES data are from humans from 12 to 71 years old with normal plasma osmolality [2].

Most of the countries in the world are facing water stress either physically or economically [1]. One of the reasons is water pollution. Metals such as copper can be found in water resources such as seas, rivers, and lakes. The main sources of environmental copper release are industries producing products such as wire, pipe, and sheet metal and the mining, smelting and refining of copper [3]. However, copper can enter the body via drinking water through household plumbing systems [3].

Copper is a trace metal that plays an important role in the human body. It can be found in dietary supplements and some foods such as organ meats, shellfish, wheat-bran cereals, whole grain products, and chocolate. The human body needs copper for making red blood cells and maintaining nerves cells and the immune systems. Copper can enhance collagen production in the body, help it to absorb iron, and contribute to the production of energy [4]. Too little or too much copper can damage brain tissue. The upper limit for adults over the age of 19 is 10 mg per day, and intake above this level could be toxic. Excessive copper consumption can lead to several diseases, especially Wilson’s disease and Alzheimer’s disease. Copper ions significantly impact the viscera and the central nervous system, especially the brain [5].

Until now, various methods have been established to measure the concentration of Cu^2+^ ion at trace levels, which include flame atomic absorption spectrometry [6], surface-enhanced Raman spectroscopy [5], atomic absorption spectrometry (AAS) [7], inductively coupled plasma atomic emission spectrometry (ICP-AES) [8], and inductively coupled plasma optical emission spectroscopy (ICP-OES) [9]. Although some of these methods offer a very good detection limit and multielement analysis, these techniques have some drawbacks in the accurate determination of Cu^2+^ detection, including costly equipment facilities, experience and well-trained personnel, and time-consuming and complex sample preparation [10,11]. Thus, a fast, simple, portable, and easy-to-use sensor for Cu^2+^ detection is urgently needed.

Optical chemical sensors have attracted many researchers due to their cost-effectiveness [10], compactness, and miniaturizability [12], and their results can be seen by the naked eye. Various methods have been proposed for detecting Cu^2+^ ions, including the fluorescence method [13], the colorimetric method [14], the surface plasmon resonance method [15,16] and reflectometry methods [11]. Although fluorescence and colorimetric methods demonstrate high sensitivity for Cu^2+^ ion detection, these methods require color-complexing, more chemical reagents, a complex experimental procedure, high-cost equipment, and a long detection time [5,15]. While the surface plasmon resonance method requires expensive metal materials, such as gold, the process of the sensor is complicated [15]. However, optical chemical sensors based on reflectance spectrometry have been shown to be sensitive in detecting a small amount of copper ion and provide a reagentless measurement, rapid response time, and less interference analysis [11].

Tetraaza compounds (TL) appear to be attractive because of their unique properties. Tetraaza is a macrocyclic compound containing a secondary amine group (N-H) and C=N that enables the formation of coordinative bonds between ligand donor and metal acceptor. The cation tends to bind with the nitrogen atom at the C=N functional group, while the anion tends to bind with the N-H functional group via hydrogen bonding. Four nitrogen atoms in this compound allow it to be an active ligand and able to bind with transition metals. This characteristic of tetraaza makes it possible to act as an ionophore, especially in metal sensing [17]. Promising research has been conducted on 5,12-dimetil-7,14-diphenyl-1,4,8,11-tetraazacyclotetradeca-4,7,11,14-tetraena ligand by adsorbing heavy metals from the river [18].

Choosing the proper immobilization site is crucial in the fabrication of chemical sensors. A suitable matrix can enhance the sensitivity of the developed sensor, since they ensure the immobilized compound remains on the matrix and allow the reaction to take place with the specific target. Mesoporous silica, namely Santa Barbara Amorphous (SBA-15), has caught the interest of researchers due to uniform hexagonal pores with clear structures, bigger pore size (4.6–30 nm), large pore volume, thick pore wall, high thermal, and hydrothermal stability [19,20,21]. Its inner surface area is more than 1000 m^2^g^−1^. Its pores have a functional group of silanol (Si-OH), which makes it possible to be modified with organic groups such as imidazole, tritiana [22], phosphoryl [23], and cyclam [24]. The thickness of the inert wall frame is approximately 3.1 nm to 56.4 nm, which contributes in hydrothermal and mechanical stability, nontoxicity, biocompatibility, and good catalytic ability [25,26].

Because of the large surface area provided by SBA15, we have studied the potential of this material for improving the performance of Cu^2+^ ion optical sensors. In this work, an optical sensor for Cu^2+^ ion detection was fabricated by immobilizing 5,5,7,12,12,14-hexamethyl-1,4,8,11-tetraazacyclotetradeca-7,14-dienium dibromide tetraaza compound (TL) onto SBA15 to form a pearl-colored substrate. TL provides a large number of binding sites for Cu^2+^ ions and acts as an ionophore. Cu^2+^ ions bind with immobilized TL via C=N and N=H functional groups and changes the color of SBA15 from pearl to violet. Cu^2+^ ion concentration was quantified based on light reflection transduction with a reflectance spectrophotometer. The combination of the tetraaza compound and SBA15 will enhance the sensitivity and the selectivity of the newly constructed Cu^2+^ ion optical sensor.

## 2. Materials and Methods

All the chemicals and reagents obtained were analytical grade and used without any purification. Ammonium bromide from Schimidt, ethylenediamine from Fluka, and acetone from Schimidt were used for the synthesis of tetraaza ionophore. Tetraaza solution was prepared by dissolving 153.1 mg of the as-prepared tetraaza compound in 4 mL aqueous acetonitrile (QReC, Selangor, Malaysia) medium (1:1 *v*/*v*). Chromium(III) nonahydrate nitrate (Aldrich, St. Louis, MO, USA), Stannous(II) chloride dihydrate (BDH, Bristol, UK), palladium(II) chloride (Systerm, Selangor, Malaysia), mercury(II) cyanide (BDH), lead(II) nitrate (Aldrich), cobalt chloride (UNILAB, Mandaluyong, Philippines), nickel(II) chloride hexahydrate (Schmidt), zinc chloride (R&M, Wetzikon , Switzerland), manganese(IV) acetate tetrahydrate (Aldrich), and copper acetate hydrate (Schmidt) were used to study the binding of tetraaza and metal ions. Acetate buffers of 1.0 M were prepared using glacial acetic acid and sodium acetate from Sigma Aldrich (St. Louis, MO, USA). The tetraaza ligand solutions were prepared using solvent acetonitrile (QReC), while the copper solutions were prepared using acetate-buffer acetonitrile (QReC) medium (7:3). The regenerative solution, which was sodium hydroxide (NaOH), was obtained from Systerm.

### 2.1. Synthesis of SBA15 and Tetraaza Compound (TL)

Mesoporous silica SBA15 was synthesized according to Zhao’s method [27]. Tetraaza compound (TL) was synthesized following the method reported by Ismail et al. [28]. Ammonium bromide (4.987 g, 0.05 mol) was reacted with ethylenediamine (3.005 g, 0.05 mol) in a 1:1 ratio using 150 mL acetone. The solution was refluxed for 2 h. In the meantime, ethylenediamine was dropped slowly into the round-bottomed flask. After 2 h, the mixture was filtered and left to evaporate for 24 h. White solid product was formed after a day. The product was collected by filtering and washing with acetone. The yield was 82% and the melting point 119–123 °C. The analysis is as follows: ^1^H RMN (400 MHz, d_4_- MEOH) δ 3.08 (NH_2_), 3.70 (s, 2H, CH_2_-5, CH_2_-5′), 3.44 (t, J = 4 Hz, 2H, CH_2_-4, CH_2_-4′), 2.81 (s, 2H, CH_2_-3, CH_2_-3′), 2.07 (s, 3H, CH_3_-2, CH_3_-2′), 1.49 (s, 6H, CH_3_-1, CH_3_-1′); ^13^C RMN (101 MHz, DMSO) δ 175.8 (C7, C7′), 58.4 (C4, C4′), 46.9 (C6, C6′), 43.7 (C5, C5′), 42.4 (C3, C3′), 23.4 (C2, C2′), 20.5 (C1, C1′). IR (cm^−1^): 1664 ν(C=N), 1225 ν(C-N), 3459, 3400 ν(NH_2_), 2975 (C-H). This characterization is according to literature [28].

### 2.2. UV-Vis Solution Study and Binding Titration

Colorimetry is a chemical analysis method based on the color similarities between sample and ligand solutions, using a polychromatic light source with an eye detector. Transition metal salts such as chromium(III) nonahydrate nitrate, stannous(II) chloride hydrate, palladium(II) chloride, lead(II) nitrate, mercury(II) cyanide, cobalt chloride, nickel(II) chloride hexahydrate, zinc chloride, manganese(IV) acetate tetrahydrate, and copper acetate hydrate were selected to complete this study. For colorimetric detection of metal ions, stock solutions of tetraaza and metal ions at 8 × 10^−2^ M were prepared separately in acetonitrile solvent (volume ratio of acetonitrile to water, 1:1). The color changes of the mixture were observed upon the addition of each set of metal ions into separate vials of tetraaza chemical reagent in a volume ratio of 1:1. The absorbance spectra were recorded on a Shimadzu UV 1800 spectrophotometer. For UV-vis binding titration study, stock solutions of TL (8 × 10^−2^ M) and a series of transaction metal ions such as Cu^2+^, Ni^2+^, Cr^3+^, Co^2+^, and Pd^2+^ (8 × 10^−2^ M) were prepared. 30 µL of TL was placed in each vial followed by the addition of 3 µL of metal ions. Each aliquot addition of 3 µL of metal ion solution were added to separate vials. The binding energy between TL and metal ion was calculated using the Benesi–Hildebrand (B–H) equation. A molar ratio graph was plotted to examine the complex stoichiometry formed between TL and the metal ions.

### 2.3. Fabrication of Optical Cu^2+^ Ion Sensor

The reflectance sensor was fabricated by depositing approximately 10 mg SBA15 into 500 µL Eppendorf round cap (diameter = 5 mm, height = 3 mm) followed by dropping 30 µL of tetraaza compound (1 mM, 0.00478 g) and left overnight to allow the TL compound completely immobilized onto the SBA15. Then, the unbound TL compound was rinsed several times using acetate buffer (pH 5) and left to dry for 30 min. About 30 µL of Cu^2+^ ion solution was loaded onto the fabricated Cu^2+^ ion sensor’s surface. The reflectance intensity of the constructed Cu^2+^ ion sensor before and after reaction with Cu^2+^ ion was measured with a fiber optic reflectance spectrophotometer. The stepwise process for the fabrication of the optical Cu^2+^ ion sensor is shown in Figure 1.

### 2.4. Characterization and Performance Evaluation of Optical Cu^2+^ Ion Sensor

The constructed optical Cu^2+^ ion sensor was optimized along several parameters including pH, TL concentration, linear range, response time, reproducibility, regeneration, shelf life, and interference, and the measurements were conducted in triplicates, respectively. The effect of pH on the Cu^2+^ ion sensor response was studied by adjusting the 0.05 M acetate buffer pH between pH 3 and pH 8. The concentration effect of tetraaza compound (TL) was conducted by changing the TL concentration from 2 mM to 10 mM at pH 6. Five mg of SBA15 was immersed for 24 h in 3 mL of TL (2 mM, 6 mM, and 8 mM) to confirm the amount of immobilized TL via UV-vis spectrometer analysis. The dynamic linear range of the sensor was determined with a series of Cu^2+^ ion concentrations from 1 × 10^−7^ M to 2 × 10^−5^ M with constant TL concentration at 6 mM and pH 6. A reproducibility test was conducted by evaluating the reflectance intensity of the eight independent Cu^2+^ sensors of the same batch by using 1 × 10^−5^ M of Cu^2+^ ion with pH 6. A repeatability test was conducted by alternately incubating the constructed optical sensor in 1 × 10^−5^ M of Cu^2+^ ion with pH 6 and 0.1 M EDTA regeneration solution (10 min) with the reflectance measurement taken after each incubation. An interference study was performed by preparing a series of potential interference ions, such as Ni^2+^, Cr^3+^, Co^3+^, V^5+^, Ir^2+^, Fe^2+^, Rh^3+^, and Pd^2+^, at different molar concentration ratios (1:0.1, 1:1, 1:10, 1:100) of 1 × 10^−5^ M Cu^2+^ ion with interfering ions. Significant deviation in the reflectance response resulting from the presence of interfering ions was estimated based on ±5% deviation from the relative reflectance intensity at 631.1 nm when no interfering ion was present in the determination of 1 × 10^−5^ M Cu^2+^ ion by the constructed optical sensor.

### 2.5. Real Samples

The fabricated optical sensor for Cu^2+^ ion determination was then validated and compared against the standard method; inductively coupled plasma-mass spectrometry (ICP-MS) technique. Five different types of teabag samples from Cameron Highlands, Malaysia have been used in this analysis. The teabag sample was immersed into a mixture of 10 mL nitric acid (0.1 M) and 90 mL distilled water and heated at 80 °C for 20 min until a yellowcolored solution appeared. Nitric acid is a strong mineral acid that produces soluble salt and is able to retain the element of interest in the solution. A series of known concentrations of Cu^2+^ ion from 6 × 10^−7^ M to 6 × 10^−6^ M were spiked into the teabag samples before being tested by the constructed optical sensor, followed by validation with the ICP-MS standard method.

## 3. Results and Discussion

### 3.1. Metal Screening and UV-Vis Binding Titration Study

Colorimetric sensing of unimmobilized TL towards a series of metal ions such as Cu^2+^, Mn^2+^, Zn^2+^, Ni^2+^, Co^2+^, Hg^+^, Pb^2+^, Pd^2+^, Sn^2+^, and Cr^3+^ was carried out qualitatively by observation by the naked eye. TL exhibited ionochromism behavior towards Cu^2+^, Ni^2+^, Cr^3+^, Co^2+^, and Pd^2+^ metal ions (Figure 2a). Obvious and instant color changes were observed upon the addition of TL to Cu^2+^, Ni^2+^, Cr^3+^, Co^2+^, and Pd^2+^ metal ions solution.

The UV-vis spectroscopy method has been used to identify the chemical reaction between unimmobilized TL towards Cu^2+^, Mn^2+^, Zn^2+^, Ni^2+^, Co^2+^, Hg^+^, Pb^2+^, Pd^2+^, Sn^2+^, and Cr^3+^ metal ions. According to Zhang et al. [29], the addition of metal ion onto a compound will form a new peak and change the absorbance value. This new peak appeared after the addition of metal ions which correspond to the d-d transition of the complexes formed. The d-d transition involves the excitation of an electron from a lower d orbital to the higher level of the d orbital. This situation occurred upon the complexation in which the energy levels of the orbital are changed accordingly and tally with the shape of the complexes. The Cu^2+^ ion appeared to be the highest absorption peak after the addition of unimmobilized TL (Figure 2b). This reveals that the unimmobilized TL showed the highest affinity towards the Cu^2+^ ion as compared to other metal ions.

UV-vis spectroscopic titration was carried out to determine the binding behavior of the TL–Cu^2+^ ion reaction in acetonitrile medium. In addition, the stoichiometry ratio of the TL–Cu^2+^ complex was obtained by gradually addition of 3 µL 1 × 10^−2^ M of Cu^2+^ ion solution into 3 µL 1 × 10^−2^ M of TL solution until no change could be seen on the UV-vis spectrum (Figure 2c). The spectrum was recorded after each addition of Cu^2+^ ion solution. Based on the UV-vis titration studies (Figure 2c), the binding constant (K*_a_*) was calculated using a Benesi–Hildebrand plot of linear regression of (1/ΔA) versus 1/[G]_0_, where A is the absorbance of the metal ion and [G]_0_ is the concentration of Cu^2+^ ion. The linear correlation observed from the graph indicates that the TL compound binds with Cu^2+^ metal ion in a 1:1 stoichiometry. The K*_a_*, binding constant value calculated from the value of intercept/slope is 2.33 × 10^3^ M^−1^ for Cu^2+^ ion. The Cu^2+^ ion will be used for further investigation because it demonstrated a vibrant color change and stronger affinity towards TL compared to other metal ions.

### 3.2. Characterization of Cu^2+^ Ion Sensor

A few parameters, such as FTIR and FESEM, have been selected to observe the characteristic modification between TL and SBA15. The features of SBA15 before and after immobilization of TL were characterized using infrared spectroscopy as shown in Figure 3a. SBA15 is represented by blue spectrum. Before the immobilization process, the FTIR band at 3410 cm^−1^ showed the presence of a stretching vibration band for the hydroxyl group (O-H) in the silanol group (Si-OH), and 950 cm^−1^ could be assigned as the organosilanes group, either Si-OH- or Si-O-stretching vibrations of the mesoporous structure [30,31]. There were symmetrical and asymmetrical stretching vibration bands at 1060 and 800 cm^−1^, which were due to the Si-O-Si network in SBA15. The broad band at 3410 cm^−1^ with weak intensity is due to the stretching vibration mode of the hydroxyl group of Si-OH, while the tiny absorption band at 1579 cm^−1^ might be due to the deformation vibration of physically adsorbed molecular water [27,32].

The green spectrum represents the TL compound. A strong adsorption band has been observed around 1652 cm^−1^, and it shows the C=N group, while 1220 cm^−1^ shows the stretching of the C-N group. Both adsorption bands exist as a result of reaction between ethylenediamine and ammonium bromide to form the TL compound. On top of that, two sharp peaks appear at 3458 and 3396 cm^−1^, which were due to the amine primer group, υNH. Adsorption band at wavenumber 2894, 2952, and 3000 cm^−1^ show the υN-H stretching group and C-H binding vibration mode. Adsorption bands at 2894, 2952, and 3000 cm^−1^ show the υC-H stretching group and C-H vibration binding mode.

After TL was immobilized onto SBA15 (red spectrum), the OH stretching band (~3410 cm^−1^) of SBA15 becomes broader at a wavenumber of 3000–3500 cm^−1^ due to the interaction between the hydroxyl group from SBA15 and the NH group from TL. The imine group C=N stretching band at 1501 cm^−1^ of TL becomes stronger and clearer, reaching 1522 cm^−1^, and the hydroxyl peak become broader because of the hydrogen bonding interaction between the nitrogen of the imine from TL with the hydrogen of SBA15 [32]. There was a hydrogen-bonding interaction between the nitrogen of the imine with the hydrogen of SBA15, the so-called hydrophobic force. The peak was around 1622 cm^−1^, indicating the presence of a C-N bond. Furthermore, the presence of adsorption at wavenumber 890, 1050, and 800 cm^−1^ indicates the existence of a Si-O organosilanes stretching group. All the data are supported by previous research [23,33].

The morphology and size distribution of SBA15 and TL–SBA15 was examined via field emission scanning electron microscopy (FESEM). It is important to confirm the size and morphology of SBA15 and TL–SBA15, as they are the main key to enhancing the sensitivity of the sensor. FESEM supplies a very clear image of the structure of the constructed Cu^2+^ ion sensor. The FESEM micrograph (Figure 3b) shows the structure of SBA15. It shows a clear image of a cylindrical shape with uniform pore diameter of 200 nm. Short cylindrical shapes and large pores will provide a superior immobilization site for TL. The inner and outer space of SBA15 increased the binding surface area, in addition to a high TL-immobilization capacity, so it turned out to be a sensitive sensor. The shape and large pores were found to influence the dispersion of molecules and ions and enhance the chances of interaction between TL and Cu^2+^ ions [24].

The FESEM micrograph showed the existence of small particles from TL on SBA15 surfaces (Figure 3c). These particles are bright and have irregular shapes. From Figure 3b and 3c, we can see that the shape of SBA15 is unchanging, but SBA15′s wall size is different in both figures. The cylindrical shape of SBA15’s wall became thicker after the immobilization process with TL (Figure 3c). This is probably due to the fact that TL was successful immobilized onto SBA15. The optimization of TL loading will be discussed to verify that TL is successfully immobilized onto SBA15. We now proceed to the sensor’s response analysis for further investigation.

### 3.3. Optical Sensor Based on TL-SBA15

SBA15 has been selected as a TL-compound-immobilizing material. TL was physically immobilized onto SBA15 via hydrophobic forces. SBA15 is categorized as an opaque material, which means that it reflects, scatters, or absorbs all light. A reflectance spectrometry transducer is the best way to measure the reflectance light intensity reflected from the opaque material’s surface. The reflectometry transducer quantifies the color intensity and reflects greater reflectance response with bright colored objects compared to a dark background substance [34]. TL–SBA15 is pearl in color before reaction with Cu^2+^ ions. During the reaction between immobilized TL and Cu^2+^ ions, the Cu^2+^ ion binds to all four nitrogen atoms of TL and exhibits a violet background color. Cu^2+^ ions have the ability to coordinate strongly and form strong coordinate bindings with compounds containing nitrogen atoms [35]. It has been observed that the clear color change of the immobilized TL after reaction with Cu^2+^ ions is similar to that in free solution, and no leaching was observed during the experiment. Figure 4a shows the reflectance spectra of the Cu^+2^ ion based on TL–SBA15. From the sensor response, it was confirmed that TL is immobilized onto SBA15. The immobilized TL showed maximum reflectance intensity at a wavelength of 630.9 nm due to the light color of TL–SBA15 (pearl in color), which has better light reflection characteristics compared to the darker surfaces. After TL–SBA15 binded with the Cu^2+^ ions, the optical sensor showed a reduction in the reflectance response at maximum wavelength (630.1 nm) due to the darker background (violet in color), which has light-absorbing characteristics. The color changes from pearl to violet drastically alter the relative intensity of the reflectance response. The Cu^2+^ ion sensor showed the largest reflectance difference at a wavelength of 631.1 nm before and after forming a complex with the Cu^2+^ ion. For that reason, 631.1 nm was chosen as the working wavelength for the entirety of the constructed Cu^2+^ ion sensor optimization experiments.

### 3.4. The Optimization of pH and TL Loading

Optimization of pH is crucial to attain a suitable pH environment for the optimum reaction to take place between TL–SBA15 and the Cu^2+^ ion. Acidic or alkaline conditions may affect the binding interaction between the sensor and Cu^2+^ ions in terms of ion adsorption, ionization, and sensor adsorbance. The optimum pH was determined based on the highest relative reflectance at the wavelength of 631.1 nm. Figure 4b shows the Cu^2+^ sensor’s relative intensity response to different pH values of the acetic buffer, from pH 3 to pH 8; pH 6 has the highest relative reflectance intensity. The intense violet color may vary depending on the pH environment. Based on observations, the reaction between TL–SBA15 and Cu^2+^ ions favored more acidic conditions as the relative intensity increased from pH 3 to pH 6. In an acidic environment, Cu^2+^ ions more favorably bind with the active sites of the immobilized TL–SBA15. The relative intensity response of the constructed sensor depends on the pH strength of the solution, as it will cause the protonation or deprotonation of the binding sites, such as the C=N and N=H groups. A pH of 6 is selected as the optimum pH based on the highest relative reflectance signal. The relative reflectance started to decrease in higher pH environments due to the hydrolysis of Cu(OH)_2_ and reduced the actual Cu^2+^ ion concentration in the reaction medium [11]. This weakened the binding interaction between Cu^2+^ ions and the immobilized TL–SBA15 ligand, and as a result, no Cu^2+^ complexes were formed. At pH levels less than 3, there is no notable color change due to the competition between H_3_O^+^ ions and Cu^2+^ ions towards binding with TL–SBA15; again, no Cu^2+^ complexes were formed [11,35].

SBA15 was selected as the TL carrier matrix. This type of matrix provides a large capacity of sites for TL immobilization. The sensitivity of the fabricated Cu^2+^ ion sensor towards Cu^2+^ ion concentration depends on the immobilization method and the optimum amount of TL immobilized on the matrix surface. The concentration of TL loaded onto SBA15 was optimized from 2 to 6 mM, as the SBA15 provided a large surface area for immobilization of a higher amount of TL (Figure 4c). Therefore, the reaction between TL and Cu^2+^ ion was high. As previously mentioned, it has proven that TL is successfully immobilized onto SBA15. The optimum sensor response was shown at a TL concentration of 6 mM and declined at 8 mM onwards. In a state of saturation, the surface of SBA15 becomes densely packed with a layer of immobilized TL, and the TL–SBA15 becomes less homogenous and produces a poorer signal [34]. UV-vis spectrometer analysis was performed to corroborate the robustness of the system. From the absorption spectra (Figure 5), the absorbance of TL increased after an overnight interaction with SBA15. This confirms that the adsorption of TL by SBA15 was good, as evidenced by the high percentage of adsorption efficiency. This indicates that adsorption or interaction occurs at the surface of SBA15.

### 3.5. Analytical Performance of Cu^2+^ Ion Sensor

The reflectance response of the developed sensor towards different molar concentrations of Cu^2+^ ion (1 × 10^−8^ to 1 × 10^−2^ M) at pH 6 was observed to construct a linear calibration curve. The incremental concentration of Cu^2+^ ion (1 × 10^−7^ M to 2 × 10^−5^ M) caused the decline of the reflectance intensity due to the color change from a pearl to violet color of the Cu^2+^ complex (Figure 6a). Figure 6a (inset) represents the linear response range of the developed Cu^2+^ ion sensor from 1 × 10^−7^ M to 1 × 10^−5^ M with a correlation coefficient of R^2^ = 0.99. The linear response curve started to plateau when the concentration of Cu^2+^ ion was more than 1 × 10^−5^ M. The plateau curve was attributed to the unchanged color of the complex and the unavailability of active sites on the microsphere surface of TL–SBA15 for further reactions with the excess Cu^2+^ ions. The limit detection of the Cu^2+^ ion sensor was calculated as three times the standard deviation of blanks divided by the linear calibration slope. Limit detection is the lowest concentration of Cu^2+^ ion that can be detected, and the limit detection (LOD) of the developed sensor was 1.02 × 10^−7^ M. The high sensitivity of the developed sensor was obtained because of large pore volume and cylindrical structure of SBA15, which offered an effective channel for Cu^2+^ ions to transmit and bind with the immobilized TL to form the Cu^2+^ complex.

In theory, the sensor’s reproducibility is hard to achieve because of constraints such as concentration of the compound, particle size, and sensor design influencing the sensor’s reproducibility [36]. Reproducibility of the sensor response of the Cu^2+^ ions has been studied by taking eight measurements using different batches of TL–SBA15 at a 1 × 10^−5^ M concentration of Cu^2+^ ion (Figure 6b). The RSD of the reproducibility of the fabricated sensor was calculated at 0.47%. From the result, it showed that the constructed sensor has an excellent reproducibility property. It indicates that this sensor is stable and able to give a consistent response.

Regeneration research was done to investigate the ability of the constructed sensor to be used several times. Regeneration time can be described as the time capture for the sensor to reach the baseline response after being immersed in the regenerating solution [37]. The regeneration performance of the Cu^2+^ ion sensor is presented in Figure 6c. Figure 6c indicates that the fabricated sensor response remained after five successive binding and generation cycles of the sensor. The response of the sensor declined after it had been soaked in 0.1 M EDTA regenerating agent for 10 min. EDTA is a weak acid, but it can break up the covalent bond between the Cu^2+^ ion and TL compound, which is two nitrogen of C=N and two nitrogen of N-H. However, the Cu^2+^ ion sensor’s relative reflectance response increased after reaction with 1 × 10^−5^ M Cu^2+^ ion solution, and the sensor response can be generated about 5 times, with relative standard deviation RSD = 0.79%. After the fifth reversible cycle, the Cu^2+^ ion sensor response decreased by 18% from the initial response. This is due to the decomposition of TL–SBA15 and hindered the production of Cu^2+^ ion–TL–SBA15 complex.

### 3.6. Selectivity of TL–SBA15 Optical Sensor for Cu^2+^ Ion Detection

Selectivity or interference study is important for fabricating a sensor that can detect specifically the Cu^2+^ ion. Some metal ions such as Ni^2+^, Cr^3+^, Co^2+^, Pd^2+^, Fe^2+^, Ir^3+^, and V^5+^ ions have been used to investigate the ability of the fabricated sensor in Cu^2+^ ion detection. Ni^2+^, Cr^3+^, Co^2+^, and Pd^2+^ ions have been chosen because they showed obvious color change after the addition of TL in the colorimetric test, while Fe^2+^, Ir^3+^, and V^5+^ ions have the potential to be found in water. The interference effect of the potential foreign species was measured at 1 × 10^−5^ M of Cu^2+^ ion, at Cu^2+^ ion: interferent molar concentration ratio of 1:0.1, 1:1, 1:10, and 1:100. Significant deviation in the reflectance response resulted by the presence of the interfering ion was estimated based on ±5% deviation from relative intensity when no interfering ion was present in detection of 1 × 10^−5^ M of Cu^2+^ ion.

Positive interference arises when the interference ion reacts with compound and produce products with higher color intensity, while negative interference results showed incomplete reaction between ion and analyte. Table 1 showed significant positive interference on the Rh^2+^ ions at 1:100 molar concentrations ratio with 15.98% of yield error. Rh^2+^ ion gives a higher relative intensity response compared to the Cu^2+^ ion sensor without any additional foreign ion. Intense grey complexes are produced when Rh^2+^ ion was present in the reaction between Cu^2+^ ion and TL–SBA15. This influenced the background color of the immobilized sensor to weaken the reflected light actively.

On the other hand, Pd^2+^, Ni^2+^, and Co^2+^ ions presented significant interference at a concentration ratio of 1:0.1. This is in line with Pearson’s principle of hard and soft acid-base (HSAB), whereby the immobilized TL acted as a soft base and had the most intense interaction with mild to soft acidic ions, including metal ions such as Pd^2+^, Ni^2+^, and Co^2+^ ions. This is driven by a simple electron transfer effect and caused competitive binding of the foreign metal ion with target Cu^2+^ ion for binding sites at TL [38].

However, this result is still acceptable. According to the National Water Quality Standard [39], Rh^2+^, Pd^2+^, and Co^2+^ ions should not be present in livestock drinking water, while the limits on concentration for Ni^2+^ ions in drinking water is 3.4 × 10^−4^ M, and the interference only occurred at 1 × 10^−6^ M of nickel concentration. Therefore, the interference effect from those potential ions can be neglected upon detection of Cu^2+^ ion with the developed optical sensor based on TL–SBA15. Otherwise, ethylenediaminetetraacetic acid (EDTA) is a widely used chelating agent that can be used to form complexes for metal ions in a simple binary mixture or a more complex mixture [40].

### 3.7. Validation Study of Cu^2+^ Ion Detection Using TL–SBA15 Optical Sensor in Teabags

The performance of the developed Cu^2+^ ion sensor was validated with the ICP-MS conventional method to determine the applicability of the developed sensor in terms of accuracy, precision, and quantification of the Cu^2+^ ion. Five types of teabags from different manufacturer were used in this study.

The teabag samples have a natural dark yellow color that could affect the performance of the developed sensor by producing a low reflectometric response compared to the actual concentration value. The teabag samples were spiked with known amount of standard Cu^2+^ ion concentration ranging from 6 × 10^−7^ M to 6 × 10^−6^ M and analyzed with the newly developed Cu^2+^ ion optical sensor and ICP-MS method as a validation method. Results from both methods were tabulated in Table 2. Based on the data, both methods showed a comparable result for Cu^2+^ ion detection in teabag samples. The calculated t values were lower than the t critical value, with 90% confidence intervals. The smaller t value showed the similarity results between the constructed Cu^2+^ ion optical sensor and ICP-MS method and indicates the constructed Cu^2+^ ion optical sensor generates comparable results to the ICP-MS standard method.

### 3.8. Performance Comparison with Other Reported Cu^2+^ Ion Optical Sensor

Many other Cu^2+^ ion sensors have been reported, but in this article, we only focused on Cu^2+^ ion optical sensors. Table 3 displayed the performance comparison of the constructed Cu^2+^ ion sensor with the other Cu^2+^ ion optical sensor in terms of linear range, limit of detection, and response time. Several Cu^2+^ ion sensors have been reported based on reflectance [35,41], fluorescence [42,43,44], and absorbance transduction with difference types of immobilization sites. The Cu^2+^ ion sensor based on TL–SBA15 showed an improvement in limit of detection and response time compared to other reported Cu^2+^ ion sensors. This is due to the large surface area for TL to immobilize onto SBA15. SBA15 has a cylindrical shape with pores that increased the TL binding capacity and enhanced the chances of interaction between TL and the Cu^2+^ ions. The superior immobilization sites of SBA15 is the key to enhancing the sensitivity of Cu^2+^ ion detection. Moreover, the implementation of TL as the sensing material will increase the binding capacity of the Cu^2+^ ion. The use of Nitroso-R reagent-sol gel has been reported to give a wider working linear range in the detection of Cu^2+^ ions [35]. However, the sensor based on Nitroso-R reagent-sol gel takes 40 min to detect Cu^2+^ ions. This newly developed reflectance Cu^2+^ ion sensor based on TL–SBA15 demonstrates a very short response time with a good detection limit.

## 4. Conclusions

A new optical sensor based on TL–SBA15 was successfully developed. This newly constructed optical sensor showed a fast response and is able to detect Cu^2+^ ions at a very low concentration down to the micromolar level. The color of TL-SBA15 changed upon the addition of Cu^2+^ ions, and it is useful for rapid and on-site detection of Cu^2+^ ions in water. The proposed Cu^2+^ ion optical sensor has high potential in developing an optical sensor kit for Cu^2+^ ion detection either in tea or any drinking water and for environmental field-testing purposes.

## Figures and Tables

**Figure 1 membranes-12-01152-f001:**
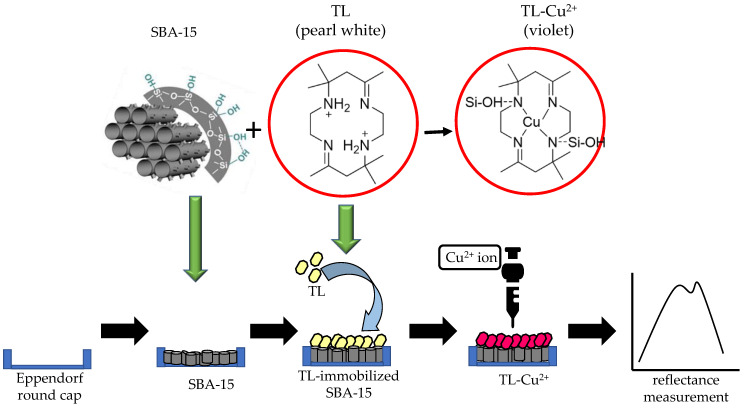
Stepwise process for the fabrication of Cu^2+^ ion sensor based on SBA15 as immobilization matrix and TL as sensing reagent. Chemical reaction of TL with Cu^2+^ ion presenting color change from pearl white to violet and quantified with reflectance measurement.

**Figure 2 membranes-12-01152-f002:**
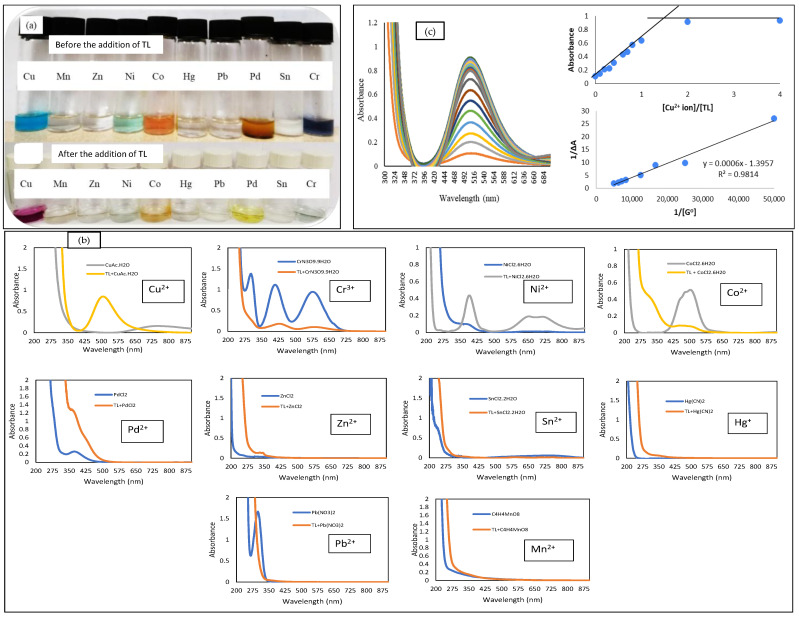
(**a**) Colorimetric sensing of TL compound towards a series of 8 × 10^−2^ M metal ions in acetonitrile solvent, before and after the addition of TL compound; (**b**) UV-vis spectrum of TL before and after the addition of metals, and (**c**) UV-vis absorption titration of TL upon gradual increase of 3 µL 8 × 10^−2^ M Cu^2+^ metal ion solution in (CH_3_CN: H_2_O, 1:1); (**c**) (inset) Molar ratio plot of absorbance against ratio of copper ion with TL, which shows a ratio of 1:1; and (**c**) (inset) Benesi–Hildebrand plot to determine the binding constant.

**Figure 3 membranes-12-01152-f003:**
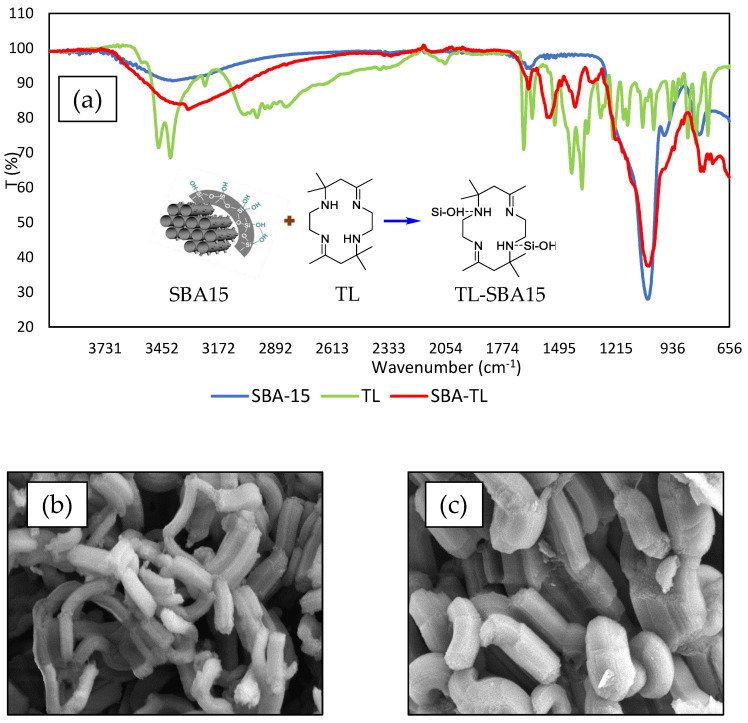
(**a**) FTIR spectra for SBA15, tetraaza compound (TL), and SBA15-TL; (**b**) FESEM micrograph for SBA15; and (**c**) FESEM micrograph for SBA15-TL at 30,000× magnification.

**Figure 4 membranes-12-01152-f004:**
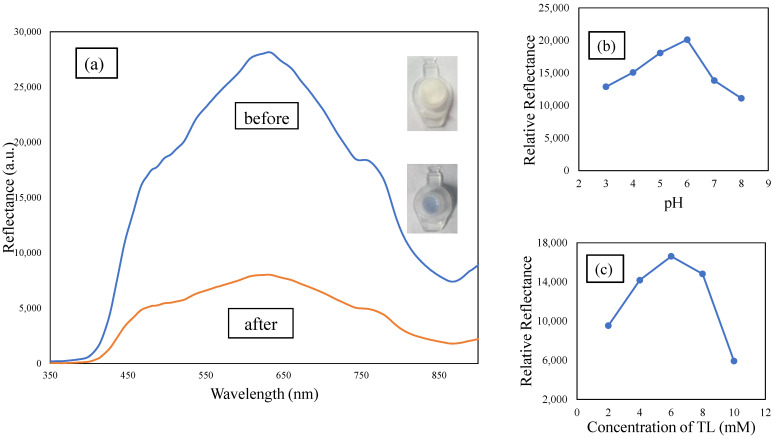
(**a**) Spectra of reflectance intensity before and after the deposition of Cu^2+^ ion onto the TL–SBA15 microsphere; (**b**) effect of pH; and (**c**) effect of TL concentration on the reflectance of the Cu^2+^ ion sensor response.

**Figure 5 membranes-12-01152-f005:**
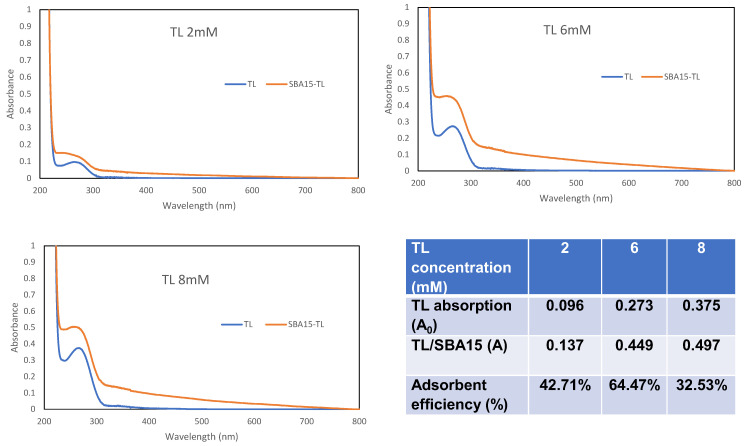
UV-vis analysis for different concentrations of TL before and after addition of SBA15.

**Figure 6 membranes-12-01152-f006:**
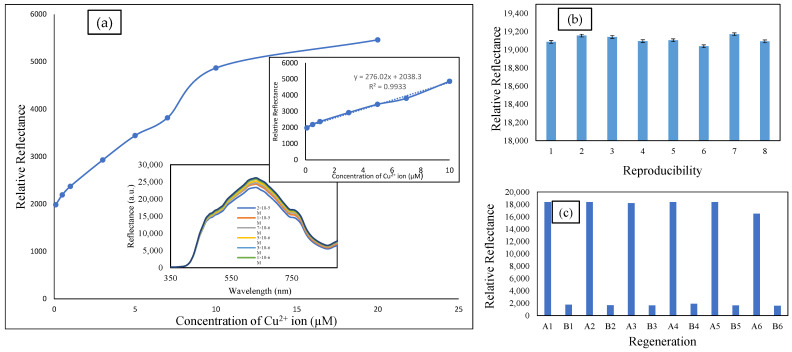
(**a**) Spectrum of the detection of Cu^2+^ ion ranging from 1 × 10^−7^ M to 2 × 10^−5^ M; (**a**) (inset) linear response range of Cu^2+^ ion detection from 1 × 10^−7^ M to 1 × 10^−5^ M; (**b**) reproducibility study of Cu^2+^ ion sensor based on TL–SBA15 (*n* = 8); and (**c**) reversibility of Cu^2+^ ion sensor with 1 × 10^−5^ M Cu^2+^ ion solution (A1, A2, A3, A4, A5 and A6) and 0.1 M EDTA regeneration solution (B1, B2, B3, B4, B5, and B6).

**Table 1 membranes-12-01152-t001:** The relative reflectance intensities obtained by the constructed optical sensor in detection of 1 × 10^−5^ M of Cu^2+^ ion in the absence and presence of potential interfering ions at pH 6 and wavelength of 631.1 nm (*n* = 3).

Interference Ion	Relative Intensity of Reflection at Ratio Concentration of Cu^2+^ Ion and Interference Ion	1	2	3	4
	1:0	1:0.1	1:1	1:10	1:100
Fe^2+^	13,164.35 ± 3.35	12,989.85 ± 1.33	13,184.10 ± 0.15	13,228.72 ± 0.49	13,354.87 ± 1.45
Pd^2+^	13,164.35 ± 3.35	11,943.23 ± 9.28 *	12,573.47 ± 4.49	12,968.34 ± 1.49	13,511.99 ± 2.64
Ni^2+^	13,164.35 ± 3.35	12,499.67 ± 4.85	12,618.28 ± 4.15	13,392.03 ± 1.73	13,822.31 ± 5.7
Ir^2+^	13,164.35 ± 3.35	12,916.99 ± 1.88	13,498.05 ± 2.53	13,647.50 ± 3.67	13,712.64 ± 4.16
Rh^2+^	13,164.35 ± 3.35	12,725.69 ± 3.33	13,410.92 ± 1.87	13,828.68 ± 4.98	15,267.87 ± 15.98
V^5+^	13,164.35 ± 3.35	12,635.07 ± 4.02	12,852.66 ± 2.37	12,916.37 ± 1.88	13,512.78 ± 2.65
Co^2+^	13,164.35 ± 3.35	12,385.15 ± 4.98	12,957.6 ± 1.57	13,795.08 ± 4.79	13,682.53 ± 3.94

**Table 2 membranes-12-01152-t002:** Validation of the constructed Cu^2+^ ion optical sensor with ICP-MS as a reference method for the detection of Cu^2+^ ion concentration in teabag samples spiked with standard Cu^2+^ ion in the concentration from 6 × 10^−7^ M to 6 × 10^−6^ M (*n* = 3).

Teabag Sample	Detection of Cu^2+^ Ion Concentration Using ICP-MS (M)	Detection of Cu^2+^ Ion Concentration Using Reflectometric TL-SBA15 Sensor (M)	t Values (t_critical_ = 2.776)
1	1.212 × 10^−6^ ± 0.00	6.51 × 10^−7^ ± 6.47 × 10^−7^	1.49
2	7.063 × 10^−7^ ± 0.00	9.90 × 10^−7^ ± 3.60 × 10^−7^	1.36
3	5.790 × 10^−7^ ± 0.00	6.15 × 10^−7^ ± 1.01 × 10^−7^	0.72
4	2.489 × 10^−6^ ± 0.00	1.47 × 10^−6^ ± 1.20 × 10^−6^	2.12
5	5.989 × 10^−6^ ± 0.00	6.42 × 10^−6^ ± 1.72 × 10^−6^	2.66

**Table 3 membranes-12-01152-t003:** Analytical performance comparison between the constructed optical sensor and several reported optical sensors for Cu^2+^ ion detection.

Immobilization Matrix	Transducer	Linear Range (M)	LOD (M)	Response Time (min)	References
TL-SBA15	Reflectance	1 × 10^−7^–1 × 10^−5^	1.02 × 10^−7^	<1	This study
Nitroso- R reagent-sol gel	Reflectance	7.87 × 10^−5^–1.57 × 10^−3^	2.22 × 10^−5^	40	[35]
1-(2-pyridylazo)-2-naphthol	Reflectance	7.34 × 10^−6^–1.56 × 10^−4^	1.88 × 10^−6^	5	[41]
Carbon nanoparticles	Fluorescence	0–30 × 10^−6^	0.44 × 10^−6^	<1	[43]
N-Bodipy	Fluorescence	2.5 × 10^−4^–1 × 10^−5^	1.28 × 10^−6^	1–5	[42]
N,K, co-doped graphene quantum dot	Fluorescence	1–5 × 10^−4^	1.32 × 10^−5^	NA	[44]

## Data Availability

Not applicable.

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
