# Peer review of "A New Sensing Material Based on Tetraaza/SBA15 for Rapid Detection of Copper(II) Ion in Water"

_membranes, 2022, doi:10.3390/membranes12111152_

Round 1

Reviewer 1 Report

The authors have developed a material based on TL/SBA15 for rapid detection of Copper(II) ion in water (tea or any drinking water). I think that the manuscript describes a promising study and can be publishing after major revision.

The manuscript should be improved with the following comments:

-        The material is not very novel since it immobilizes in a known mesoporous material SBA-15, a Tetraaza (Tl), which is also a well-known compound. My first problem is about the characterization. Some dates about Tl are described and referenced to paper 28. Both descriptions do not match, for example in the m.p. data and so on

-        On the other hand, the material TL-SBA15 would need a deeper description. The amount of immobilized TL, if produced by hydrophobic interactions, could be different in each batch. If, as the authors say, the system is robust, more data should be given to corroborate it.

-        Several cations have been studied, both in preliminary studies and on the material as interferents. I think it would be interesting to study Cu(I) and compare the results

-        Why the preliminary studies with cations are carried out in acetonitrile water mixtures?

-        In table 1, Cu(II) data of relative reflectance intensities must be added to compare

-        In table 2 of the validation of the method, the errors shown are quite high if we compare it with the LoD (that is not indicated as calculated) by the authors of 1.02 x 10-7

-        There are some typographical errors, for example MEOH instead of MeOH and subscripts in figure legends. The manuscript must be revised

Author Response

Dear Editor,

The manuscript has been revised in rebuttal to the reviewers' comments. The typographical errors have been corrected accordingly. Please find the points to points comments attached here. We hope our revised manuscript can be accepted for publication. Please refer to the attachment for the point-to-point comments. We are hopeful that our revised manuscript will be accepted for publication. Thank you

Reviewer 1

  • The material is not very novel since it immobilizes in a known mesoporous material SBA15, a Tetraaza (TL) which is also a well-known compound. My first problem is about the characterization. Some data about TL are described and referenced to paper 28. Both descriptions do not match, for example in the m.p. data and so on.
  • Thank you very much for alerting us. 5,5,7,12,12,14-hexamethyl-1,4,8,11-tetraazacyclotetradeca-7,14-dienium dibromide (TL) has been synthesized according to paper 28 (Ismail 2012). Changes has been done in reference 28.

  • On the other hand, the material TL-SBA15 would need a deeper description. The amount of immobilize TL, if produce by hydrophobic interactions, could be different in each batch. If, as the authors say, the system is robust, more data should be given to corroborate it.
  • We emphasized that there is a strong interaction between TL-SBA15 based on the sensitivity of the developed sensor. The more quantity of TL immobilized onto SBA15, the more sensitive the developed sensor.

  • Several cations have been studied, both in preliminary studies and on the material as interferent. I think it would be interesting to study Cu(I) and compare the results.
  • Thank you for your very kind suggestion. We are eagerly to add this data in our manuscript if we are given enough time to conduct the experiment.

  • Why the preliminary studies with cations are carried out in acetonitrile water mixtures?
  • The preliminary studies were performed in acetonitrile-water mixtures in order to improve the solubility of metal cation. For example, palladium(II) chloride was insoluble in water but soluble in organic solvent (acetonitrile). The incorporation of water and acetonitrile significantly improves the solubility of all the transition metal salts. Furthermore, the mixture of acetonitrile and water gives the most vibrant and intense colour changes upon colourimetric sensing of tetraaza compound (TL) towards metal ions. Thus, the solvent mixture was chosen to further the optical sensor.

  • In table 1, Cu(II) data of relatives reflectance intensities must be added to compare.
  • This data is already in the table. Data 1:0 is referred to Cu(II) ion.

  • In table 2 of the validation of the method, the errors shown quite high if we compare it with LOD (that is not indicated as calculated) by the authors of 1.02 × 10-7
  • To overcome this problem, we have done t-test to verify the data. All data are less than tcritical and we confirm that all data is significant with a good agreement with standard method (ICP-MS).

  • There are some typographical errors, for example MEOH instead of MeOH and subscripts in figure legends. The manuscript must be revised.
  • Typographical errors have been amended.

Reviewer 2 Report

The authors constructed new optical sensor based on 5,5,7,12,12,14- hexamethyl-1,4,8,11-tetraazacyclotetradeca-7,14-dienium dibromide (TL) immobilized on Santa Barbara Amorphous (SBA-15). The results showed that constructed sensor allowed to obtain a fast response for Cu2+ ion detection in teabag samples at a very low concentrations of Cu2+. The resulting materials were characterized by Field Emission Scanning Electron Microscopy and Fourier Transform Infrared Spectroscopy. The Binding study towards Cu2+ ion was performed using UV-vis spectroscopy. The resulting sensor based on Tetraaza/SBA15 demonstrated a fast response for Cu2+ determination at a very low concentration down to micromolar level. The paper is well written and is suitable for this Journal. It is acceptable for publication after the minor corrections. Authors should improve the quality of Figures: 2,4 and 5 (the numbers are hard to read). Moreover Figure 2c should be reissued in order to improve the resolution.

Author Response

Dear Editor,

Thank you very much for the comments. The quality of Figures 2 (2c),4 and 5 have been improved. Please find the attached revised manuscript. Thank you.

Round 2

Reviewer 1 Report

The authors do not reply to points 1,2,3 where there were even experiments to be carried out. In these conditions I cannot recommend your publication

Author Response

Reviewer 1

  • The material is not very novel since it immobilizes in a known mesoporous material SBA15, a Tetraaza (TL) which is also a well-known compound. My first problem is about the characterization. Some data about TL are described and referenced to paper 28. Both descriptions do not match, for example in the m.p. data and so on.
  • Yes, the material is not very novel but it is a new combination between SBA15 and TL that has never been explored. This new combination is a novelty of this research in Cu2+ ion detection.
  • 5,5,7,12,12,14-hexamethyl-1,4,8,11-tetraazacyclotetradeca-7,14-dienium dibromide (TL) has been synthesized according to paper 28 (Ismail 2012). We are really sorry because we are wrongly writing the title and publisher for reference 28. Changes has been done in reference 28. In this reference, characterization such as m.p. data are tally with our research.

  • On the other hand, the material TL-SBA15 would need a deeper description. The amount of immobilize TL, if produce by hydrophobic interactions, could be different in each batch. If, as the authors say, the system is robust, more data should be given to corroborate it.
  • In figure 4(c), we have done the experiment to identify the amount of immobilize TL and the highest amount of TL immobilized onto SBA15 is 6mM. The more quantity of TL immobilized onto SBA15, the more sensitive the developed sensor.
  • To corroborate the robustness of the system, we performed an additional experiment using UV-vis spectrometer. 5mg of SBA15 were immersed 24 hours in 3mL of TL (2mM, 6mM and 8mM ). Then the solution was analyzed by UV-vis spectrometer.

TL concentration (mM)

2

6

8

TL absorption (A0)

0.096

0.273

0.375

TL/SBA15 (A)

0.137

0.449

0.497

Adsorbent efficiency (%)

42.71%

64.47%

32.53%

*measure after overnight immobilization

*efficiency = (A0-A)/A0, where A0 and A are initial absorbance of TL and equilibrium absorbance of TL/SBA-15.

From the absorption spectra, the absorbance of TL increased upon interaction with SBA-15 overnight. This confirms that the adsorption of TL by SBA-15 was good which was demonstrated by the high percentage of adsorption efficiency. This means that adsorption or interaction occurs at the surface of the SBA-15. The finding is consistent with the previous investigation that reported SBA-15 has good adsorption capacity (Qin et al. 2013; Chaudary & Sharma 2017). The UV-vis results are tally with analysis in Figure 4(c) which revealed the optimum amount of TL is 6mM.

-This data has been added in section 3.4

  • Several cations have been studied, both in preliminary studies and on the material as interferent. I think it would be interesting to study Cu(I) and compare the results.
  • Thank you for your very kind suggestion. Normally a single charge ion could not form a complex with a ligand or compound that is selective for a double charge ion. Therefore, Cu(I) is expected not to bind with TL.
  • We did not add Cu(I) ion in our study because Cu(I) is not the main interferent ion. Besides, Cu(I) is difficult to exist naturally in the environment. In an aqueous medium, Cu(II) ion is more stable than Cu(I) ion because although energy is required to remove one electron from Cu(I) to Cu(II) ion, high hydration energy of Cu(II) compensates for it. Thus, Cu(I) ion is unstable in an aqueous solution and is easily oxidize to Cu(II) (aspirationsinstitute.com).

Round 3

Reviewer 1 Report

Although I do not agree with some of the authors' comments, I believe that the manuscript can be published. There are still some typographical errors like MEOH instead of MeOH